# Switching Intention and Behaviors to Wetland Ecotourism after the COVID-19 Pandemic: The Perspective of Push-Pull-Mooring Model

**Ying-Wei Wu [1], Ting-Hsiu Liao [1,*], Shang-Pao Yeh [2] and Hao-Chen Huang [3]**

[1] Graduate Institute of Tourism Management, National Kaohsiung University of Hospitality and Tourism, Kaohsiung 812301, Taiwan; ryanwu@mail.nkuht.edu.tw

[2] Department of Hospitality and M.I.C.E. Marketing Management, National Kaohsiung University of Hospitality and Tourism, Kaohsiung 812301, Taiwan; shangpao@ms12.hinet.net

[3] Department of Public Finance and Taxation, National Kaohsiung University of Science and Technology, Kaohsiung 807618, Taiwan; haochen@nkust.edu.tw

\* Correspondence: tiffanyups@hotmail.com

**Abstract:** This study used a push–pull–mooring model (PPM model) to build an integrated model to explain the influencing factors of tourists' switching intention to wetland ecotourism after the COVID-19 pandemic. The push effect is crowding perception, the pull effect is nature-based destination attractiveness, and the mooring effect is the risk perception of COVID-19. The study collected 551 valid research samples by questionnaire survey in two world-class wetlands in Taiwan. The results of the regression analysis showed that push, pull, and mooring influenced tourists' switching intention to wetland ecotourism. Among them, the mooring effect regulated the relationship between the push effect and switching intention to wetland ecotourism, but did not regulate the relationship between the pull effect and switching intention to wetland ecotourism. Finally, the switching intention to wetland ecotourism further influenced wetland ecotourism behaviors. It is expected that people can go outdoors after the COVID-19 pandemic and bring substantial economic benefits of tourism to wetland ecological attractions in Taiwan.

**Keywords:** ecotourism; switching intention to wetland ecotourism; wetland ecotourism behavior; push–pull–mooring model; COVID-19 pandemic



## 1. Introduction

Tourism creates considerable wealth and job opportunities for every country, but correspondingly has many negative influences, such as an increase in $CO_2$ emissions and large ecological footprints in transportation and accommodation [1]. Sustainable tour models are needed to meet the needs of increasing the incomes of the travel industry and reducing the negative influences of tours such as lowering carbon emissions. Rooted in the concept of sustainable development, sustainable tours are considered as an effective measure to reduce the negative influences of tours and to promote the balanced development of destinations [2]. Especially since the COVID-19 outbreak, commercial tourism has suffered greatly [3] as many people no longer consider crowded urban tours as their first option for travel destinations. If the danger perceived by tourists is beyond acceptable levels, it may influence their tour decision-making behaviors [4]. People's opinions on risks influence their tour choices during the COVID-19 pandemic [5,6]. Therefore, by walking in nature, ecotourism has become a new type of tour for people. Ecotourism is different from a traditional city tour, a type of sustainable tourism, and a new way of touring, as it can generate a strong tourist economy and high social and environmental benefits. During the touring process, tourists' perception of a sustainable environment and their behaviors enable them to achieve a high-quality tour experience, social benefits, environmental benefits, and economic benefits.

Seasonality is an important determinant of tourism competitiveness [7,8]. Wetlands show different natural features and environmental ecology in different seasons, which could attract tourists. A wetland is a suitable place for ecotourism. Taiwan has two world-class wetlands, Sihcao Important Wetland and Zengwun Estuary Important Wetland, and both are in Taijian National Park, which is the only wetland ecological system at the junction of a river and sea in Taiwan, with two ecological landscapes of wetland and lagoon. In particular, the park has about 1200 black-faced spoonbills (Platalea minor), far more than in any other country, making it a national park of conservation significance and with many ecological characteristics. There are many famous attractions around Taijiang National Park, including Sicao Artillery Fort, Sicao Ta Chung Temple, Mangrove Green Tunnel, Salt-Pan Eco-Village, and Luermen Matsu Temple. Ecotourism visitors who go to Taijiang National Park can visit nearby attractions, creating high economic output value for the local area.

Some scholars have dedicated their research to ecotourism [9,10]. Lee and Jan in [9] developed an ecotourism behavior scale, while in [10] they predicted the factors that may influence ecotourism behaviors by multiple theories (such as theory of planned behavior, technology acceptance model, value–belief–norm theory, and social identity theory). These studies have made considerable contributions to research on ecotourism. However, according to a literature review, there are few studies on switching intention and behavior to ecotourism after COVID-19, indicating a research gap on this topic. Specifically, this study intended to explore the reasons for tourists' switching intention and behaviors to ecotourism and the factors contributing to their switching intention to ecotourism. Therefore, this study explored the factors influencing their switching intention and behavior to ecotourism based on the migration theory of the push–pull–mooring model (PPM model) proposed by Moon [11]. A literature review and deduction of research hypotheses are conducted in the next chapter to explain the reasons why the PPM model was used as the theoretical basis for this study. Following that are descriptions of the research structure, measurement variables, research location, and sample collection method of this study. The empirical analysis includes common method variance analysis, confirmatory factor analysis, and regression analysis. Finally, research conclusions and implications are proposed.

## 2. Theory and Hypotheses

### 2.1. Migration Theory of Push–Pull–Mooring Model (PPM)

Migration refers to the movement of migrants between two places over a period of time. Population migration is influenced by push and pull, a theory which dates back to 1885 when Ravenstein proposed the Law of Migration based on the analysis and induction of observed data, including people's birthplace and place of residence.

Lee [12] observed that human migratory behaviors are influenced by push and pull, and so, established a push–pull system. The push–pull system explains the reason for people migrating. Under the conditions of a market economy with free movement of people, immigrants move because their living conditions can be improved through migration. The Law of Migration proposed by Ravenstein [13] is considered as the preliminary law that implies the influences of push and pull. Longino [14] proposed an interference factor other than push and pull for the Law of Migration, namely, the variable of mooring. Longino [14] used the term mooring to describe the influences of migrants' individual factors, such as behaviors, culture, and social identity, on their migration decisions. Moon [11] further combined the concept of mooring with the original push–pull theory to develop a push–pull–mooring (PPM) theory that uses overall and individual factors to explain population migration, and considered that mooring includes variables such as individual, social, and cultural effects, which are used to interfere with migrants' decisions. Population geographers point out that migration can be regarded as people's behaviors to change their place of residence [15]. Migration is influenced by three forces: the push of the original place of residence, the pull of the attraction of the new place of residence, and the ties of mooring, namely, the PPM theory of population migration [11].

### 2.2. Ecotourism

The term ecotourism dates back to 1965 when Hetzer [16] suggested rethinking culture, education, and tour, and advocated so-called ecotourism, which now has become the basic concept of international and sustainable conservation development. The International Ecotourism Society (TIES) pointed out that ecotourism activity should follow five principles: minimize influences; respect local environment and culture; help visitors and hosts to have a positive experience; provide economic benefits to support conservation and protect the well-being of local people; and increase visitors' sensitivity to the politics, environment, and society of a country [17]. The International Ecotourism Society gave a widely accepted definition for ecotourism in 1991: ecotourism is a type of tour with environmental responsibility, and its ultimate goal is to protect the natural environment and improve the well-being of local residents. Chiu, Lee, and Chen [18] argued that ecotourism attaches importance to the sustainable development of the environment, and environmentally responsible behaviors belong to a type of environmental protection mechanism. Additionally, Cai, Liu, and Zhang [19] pointed out that ecotourism refers to the special utilization of natural areas without disturbance and pollution, where tourists can enjoy natural activities, learn to protect local resources, and give back to community development to achieve the ultimate goal of sustainable management.

### 2.3. Ecotourism Behaviors

Ecotourism behaviors refer to environmentally responsible behaviors. In the context of ecotourism, when tourists understand the impact of their actions on the environment and adhere to the norms of ecological attractions, they will maintain environmentally responsible behaviors [20]. One characteristic of ecotourism is that tourist behaviors are beneficial for or can reduce the negative effects on ecotourism destinations. Lee and Jan [6] noted that ecotourism behaviors are environmental protection behaviors, environmentally friendly behaviors, behaviors complying with ecotourism guidelines, site-specific ecological behaviors, behaviors beneficial for socio-culture, economically beneficial behaviors, and learning behaviors.

### 2.4. Push–Pull Mooring Model and Switching Behaviors

According to Keaveney and Parthasarathy [21], customer switching behaviors mean that customers continue to use existing services, but switch from original providers to other service providers. Bansal, Taylor, and James [22] stated that it is quite clear to compare the phenomenon of consumers switching service providers to the framework of the PPM model. The present study holds that the behavior of switching from original tour destinations to other tourist destinations is similar to that of migrants migrating from their original place of residence to other places. Therefore, this study explains the behavior of switching tour intention using the PPM model.

According to the literature review, some studies have introduced the PPM model into research on behaviors [22]. Bansal, Taylor, and James [22] explored consumer behavior of switching service providers based on the PPM model proposed by Moon [11], and their studies noted that push, pull, and mooring significantly influence consumers' intention to switch service providers. There have been studies on the intention and behavior switching of tours [23–26]. For example, Jung, Han, and Oh [23] explored traveler behavior of switching airlines with 529 interviewees at international airports in South Korea as the subjects. Zhang, Oh, and Lee [26] explored why consumers discontinue using peer-to-peer (P2P) accommodations and return to traditional accommodations (namely, hotels). Xie and Luo [25] investigated the determinants of tourists' decision to return to theme parks as a result of the pandemic in 2020. Xiang, Xu, and Wang [24] looked into the relationship between tourist consumption behaviors and alternative consumption intentions in abnormal circumstances. The PPM model has been used in most past studies in relation to the network industry, telecommunications service industry, and retail industry, but has rarely been used on the intentions toward and behaviors of switching tours.

*2.5. Hypotheses*

In the PPM model, the push factor refers to factors bearing negative influences on people's living quality in their original place of residence. It stimulates people to leave their original place of residence [11]. The push effect in this study is crowding perception. The phenomenon of tourist crowding involves physics and psychology. On the physical aspect, studies on the formation of tourist crowding date back to studies on recreation carrying capacity in the 1960s. Through the measurement and calculation of the carrying capacity of a recreation area, if the number of people in the recreation area exceeds a certain value, then it indicates the existence of the phenomenon of tourist crowding [27]. The crowding phenomenon refers to a state in which the carrying capacity of a recreation area exceeds its limit.

Since the 1970s, the focus of studies has moved from spatial capacities of recreational areas to psychological perception capacities of tourists [28–30]. Manning, Valliere, Wang, and Jacobi [31] generally considered that density is the basic factor for the formation of psychological crowding, and factors such as individual characteristics, social relations, and cultural psychology are combined with density, by tourists, to form crowding perception. Since the COVID-19 outbreak, passenger flow stress caused by a large number of tourists concentrating in a scenic area in a short period of time often leads to a negative tour experience and creates certain risks to public health and safety. Past studies showed that many tourists dislike crowded destinations and try to avoid them [32]. Chan [33] stated that people's perception of natural disaster risks caused by COVID-19 changes their intention regarding travel behaviors. Therefore, this study proposes the following hypothesis.

**Hypothesis 1 (H1).** *The stronger the push effect (namely, crowding perception) that tourists feel, the stronger is their intention to switch to wetland ecotourism.*

In the PPM model, the pull factor refers to factors stimulating migrants to move to their destinations [11]. Bansal, Taylor, and James [22] pointed out in their study that the pull factor is what attracts people to move to their destinations. The pull effect in this study is the nature-based destination attractiveness. Mayo and Jarvis [34] defined destination attractiveness as "the perceived ability of the destination to deliver individual benefits". Nature-based destinations take into account all kinds of natural features in attractions. Deng, King, and Bauer [35] considered that the nature-based destination attractiveness includes (1) tourism resources, (2) tourist facilities, (3) accessibility, (4) local communities, and (5) peripheral attractions. Taiwan has two world-class wetlands with rich ecosystems and many attractions around them that are attractive to many tourists. The attraction of nature-based destinations is the pull for tourists to adopt wetland ecotourism behaviors. The pull may cause tourists to change their original tour form from an urban tour to wetland ecotourism. Therefore, this study proposes the following hypothesis.

**Hypothesis 2 (H2).** *The stronger the pull effect (namely, nature-based destination attractiveness) that tourists feel, the stronger is their switching intention to wetland ecotourism.*

In the PPM model, mooring refers to personal and social factors that may accelerate migrants to depart from or keep them in their original place of residence [11]. The push–pull effect is sometimes very strong, however, for a group of people who live in a certain area and are equally influenced by the push from the area of origin and the pull from the area of destination, some people migrate and some do not. Lee [12] pointed out that this is due to environmental constraints, and, therefore, only push–pull cannot explain this phenomenon.

The mooring effect in this study is the risk perception of COVID-19. The World Health Organization (WHO) declared COVID-19 a public health emergency of international concern (PHEIC) on 30 January 2020. Due to the high infectivity of COVID-19, people have become infected in all countries. The World Health Organization (WHO) declared

COVID-19 a global pandemic on 11 March 2020, and countries around the world developed various anti-pandemic measures to try to stop the spread of COVID-19. Although the pandemic has been controlled in Taiwan quite well, there are still some restrictions on tour activities. People believe that they are at risk of being infected with COVID-19 during tour activities, which makes them aware of the increased risks of a tour and results in a significant reduction in the number of tourists at attractions. Risks have negative influences on tours [36,37], because people will be highly cautious about and fearful of tours, if they perceive high risks of infection at their destinations [38,39]. The mooring created during the COVID-19 pandemic may cause tourists to change their tour form from an urban tour to a wetland ecotourism. Therefore, this study proposes the following hypothesis.

**Hypothesis 3 (H3).** *The stronger the mooring effect (namely, risk perception of COVID-19) that tourists feel, the stronger is their switching intention to wetland ecotourism.*

In the PPM model, during the process of making decisions, the mooring effect not only influences migrants' switching intention, but also moderates the push, pull, and migration intentions [11]. People have been more cautious about their health since the COVID-19 pandemic, thus affecting the intention to health tours [40]. Tourists will change their original intention on a tour when perceiving the crowding of attractions in cities and consider that a nature-based wetland ecology is more attractive because it is less crowded. If tourists perceive the high risks of COVID-19, then they will strongly change their intention to travel to wetlands for ecotourism. Therefore, this study proposes the following hypotheses on moderating effects.

**Hypothesis 4 (H4).** *The mooring effect (namely, risk perception of COVID-19) positively strengthens the relationship between tourists' perception of push effect (namely, crowding perception) and their switching intention to wetland ecotourism.*

**Hypothesis 5 (H5).** *The mooring effect (namely, risk perception of COVID-19) positively strengthens the relationship between tourists' perception of pull effect (namely, nature-based destination attractiveness) and their switching intention to wetland ecotourism.*

The so-called behavioral intention refers to the subjective probability judgment of individuals to take a specific behavior and reflects the willingness of individuals to take a specific behavior. Behavioral intention is the most direct determinant for individuals to decide whether to take a specific behavior or not, and considers that all factors that may influence behaviors indirectly influence behavior performance through behavioral intention. According to many studies, behavioral intention, as the best method to predict individual behaviors under certain circumstances, highly correlates to behaviors [41–43], perhaps because people tend to maintain behavioral continuity and value consistency [44].

In terms of studies on tour behaviors, Jang and Namkung [45] considered that tourists' mental states would significantly influence their future tours. Past studies on tours showed that tourist intention influenced tour behaviors [25]. In the case of strong intention on wetland ecotourism, tourists will take action and change from an urban tour to wetland ecotourism. Therefore, this study proposes the following hypothesis.

**Hypothesis 6 (H6).** *Tourists switching intention to wetland ecotourism has positive influences on their wetland ecotourism behaviors.*

### 3. Methods

#### 3.1. Conceptual Framework

Based on the push–pull–mooring model (PPM model) proposed by Moon [11], this study explored tourists' switching intention and behaviors to wetland ecotourism since the COVID-19 outbreak. An integrated model was established in this study to explain the relationship among variables. The conceptual framework is shown in Figure 1.

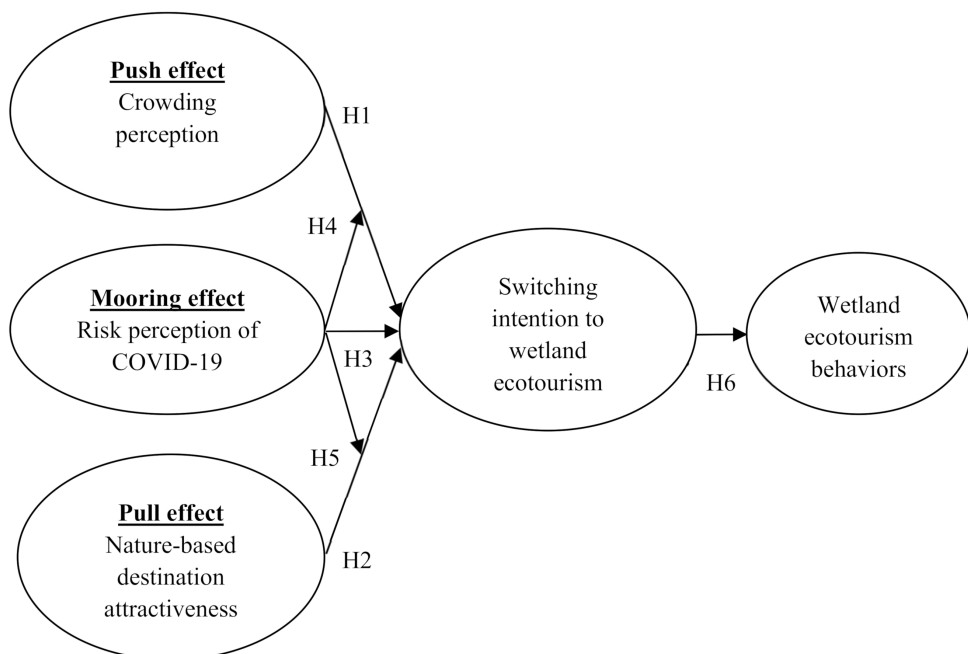

**Figure 1.** Conceptual framework.

*3.2. Measures*

Answers to items were scored on a 5-point Likert scale, ranging from 1 for "Strongly disagree" to 5 for "Strongly agree". Demographic variables were gender, educational level, marital status, family status, age, past tour experience to wetlands, intention to visit nearby attractions, and consuming behavior.

### 3.2.1. Push Effect

The push effect in this study is crowding perception. In this study, crowding perception is defined as tourists perceiving the phenomenon of crowded spaces and heavy traffic at famous attractions in cities. The following measurement items were developed according to Jacobsen, Iversen and Hem [32], Hou, Zhang and Li [46], and Luque-Gil, Gomez-Moreno and Pelaez-Fernandez [47].

1.  I think famous attractions in cities are usually densely populated;
2.  I think famous attractions in cities usually strengthen the likelihood of traffic congestion;
3.  I think famous attractions in cities often make people uncomfortable due to crowded spaces;
4.  I do not think people can maintain proper social distancing (more than 1.5 m indoors, more than 1 m outdoors) at famous attractions in cities.

### 3.2.2. Pull Effect

The pull effect in this study is nature-based destination attractiveness. In this study, nature-based destination attractiveness is defined as the degree to which tourism resources, accessibility, local communities, and peripheral attractions of wetlands attract tourists. The following measurement items were developed according to Deng, King and Bauer [35], and Reitsamer, Brunner-Sperdin and Stokburger-Sauer [48].

1.  I think the unique ecological environment of wetland attractions attracts me;
2.  I think the convenient transportation of wetland attractions attracts me;
3.  I think the featured customs of local communities at wetland attractions attract me;.
4.  I think the peripheral attractions at wetlands attract me.

### 3.2.3. Mooring Effect

The mooring effect in this study is the risk perception of COVID-19. In this study, the risk perception of COVID-19 is defined as the perceived risk of COVID-19 during a tour at

wetland attractions. The following measurement items were developed according to the studies of Chan [33], Bae and Chang [5], and Neuburger and Egger [6].

1.  I am worried about COVID-19 in areas of wetland attractions;
2.  I am concerned about the potential for COVID-19 to spread in areas of wetland attractions;
3.  I am concerned about being quarantined due to contact with people infected with COVID-19 in areas of wetland attractions;
4.  I am concerned about being infected with COVID-19 in areas of wetland attractions.

### 3.2.4. Switching Intention to Wetland Ecotourism

In this study, switching intention to wetland ecotourism is defined as the possibility of tourists switching from an urban tour to wetland ecotourism. The following measurement items were developed according to Keaveney and Parthasarathy [21].

1.  I originally wanted to travel to cities, but since COVID-19 will consider wetland ecotourism;
2.  I originally wanted to travel to cities, but since COVID-19 will travel to wetlands for ecotourism;
3.  Overall, wetland ecotourism is highly possible for me since COVID-19.

### 3.2.5. Wetland Ecotourism Behaviors

In this study, wetland ecotourism behaviors are defined as tourist behaviors that benefit or reduce negative influences on the environment, economy, and socioculture of ecological attractions of wetlands. The following measurement items were developed according to Lee and Jan [9].

1.  I will not damage the local wetland ecosystem during my trip;
2.  I will respect the local culture of wetland attractions during my trip;
3.  I will choose tour products that will not harm the local environment of wetland attractions during my trip;
4.  I will buy special local products, souvenirs, or handicrafts at wetland attractions during my trip.

### 3.3. Study Area

This study investigated two world-class wetlands in Taiwan's Taijiang National Park, including Sihcao Important Wetland and Zengwun Estuary Important Wetland. Located in Annan District, Tainan City, Sihcao Important Wetland, with an area of 551 hectares, is at the confluence of Zengwun River, Luermen River, Yanshui River, and Yanshui River Drainage (Jianan Dazhen Drainage Line) and on the southwest side of Provincial Highway 17. Sihcao Important Wetland has rich ecological resources and about 200 species of vascular plants. There are 200 species of 49 families of birds in this wetland, among which the species and number of migratory birds are the largest, accounting for about 75%. In addition, there are rare and protected species, such as black-faced spoonbills and avocets, indicating the rich and diverse bird resources in Sihcao Important Wetland. The geographic scope of Sihcao Important Wetland is shown in Figure 2.

Zengwun Estuary Important Wetland, with an area of 3001 hectares, is located in Zengwun Estuary in Tainan City, from Guosheng Lighthouse (also known as Qigu Lighthouse), the south embankment, Haipu dike, and Jiukuaicuo embankment in the north, the Qingcaolun embankment on the south bank of Zengwun River in the south, the west embankment and Provincial Highway 17 (Guoxing Bridge) to the east, and the 6 m isobath to the west. In total, 46 species of black-faced spoonbills (including 12 protected species) have been recorded in Zengwun Estuary Important Wetland. There are as many as 205 species of shellfish in this area, including edible shellfish of important economy, such as marine oysters and clams. The geographic scope of Zengwun Estuary Important Wetland is shown in Figure 3.

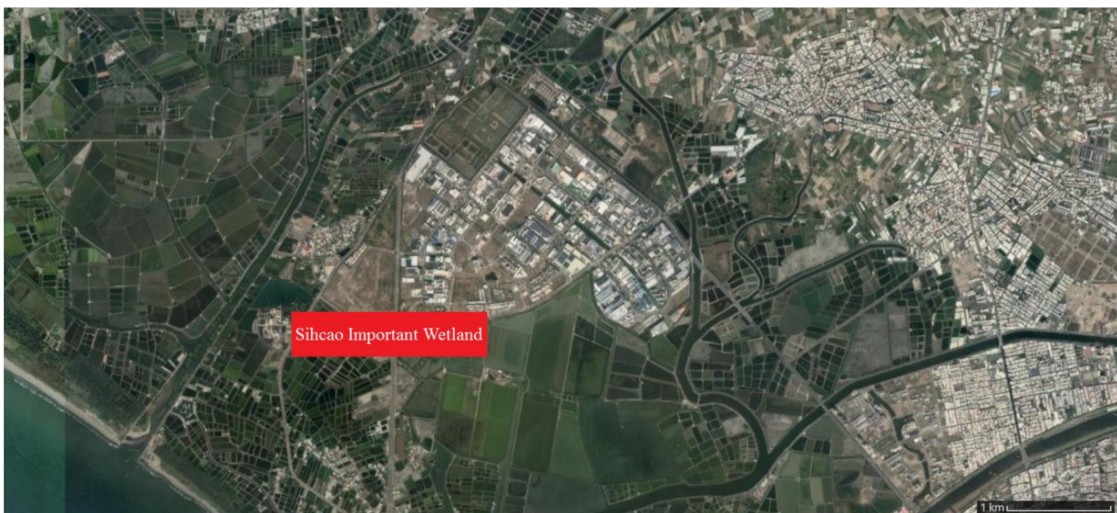

**Figure 2.** Map of Sihcao Important Wetland (from Urban and Rural Development Branch, Construction and Planning Agency, MOI).

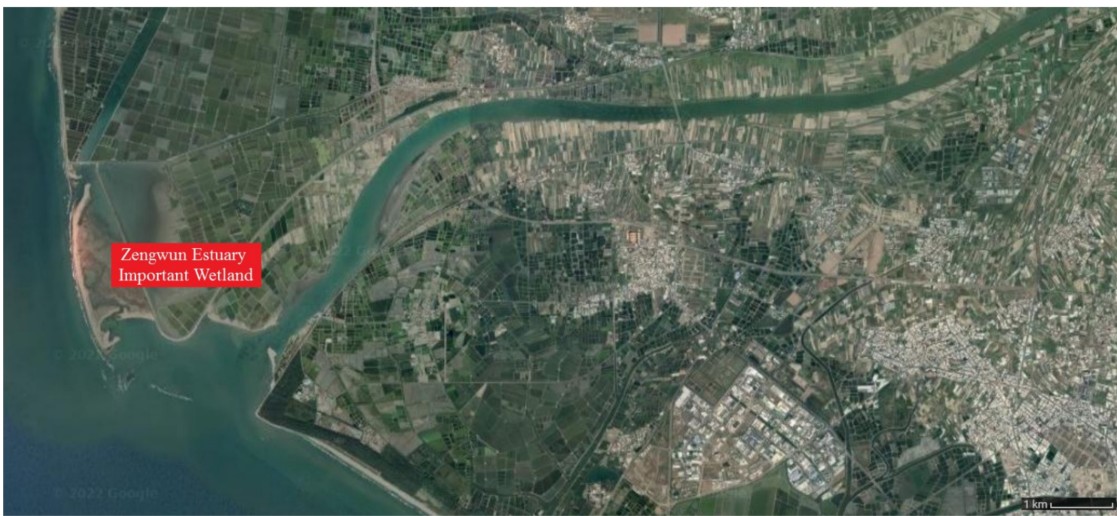

**Figure 3.** Zengwun Estuary Important Wetland (from Urban and Rural Development Branch, Construction and Planning Agency, MOI).

### *3.4. Sample and Procedure*

The study selected a total of 50 tourists who have visited Taijiang National Park, for the pre-test. In the formal questionnaire stage, convenience sampling was adopted. The formal sampling procedure occurred as follows. Questionnaires were issued at Sihcao Important Wetland and Zengwun Estuary Important Wetland in summer (July) and winter (January), to prevent the problem of underrepresentation of the sampling. The questionnaire survey was conducted in two forms. The first was the online questionnaire survey, in which online questionnaires were sent to e-mail addresses provided by tourists who were not able to complete the questionnaires on-site. The second method was the on-site questionnaire survey, in which questionnaires were provided on-site for tourists. The questionnaire interviewers received professional training and conducted the questionnaire survey without bothering tourists. Among the 600 questionnaires issued, 560 samples were collected, of which 551 were valid, for a response rate of 91.83%. The majority of the questionnaires that could not be collected were issued online. Among the valid questionnaires, 327 were issued on-site, and 224 were issued online.

The subjects of the study were tourists visiting two wetland areas, while tourists visiting non-wetland attractions or attractions in other areas were not included. The

intentions and behaviors of tourists converting their normal travel destinations to wetland tourism, the topics explored by this study, were the major concerns in the research design. The study focused on the intentions and behaviors of this conversion. Tourists visiting non-wetland attractions or attractions in other areas might not know the wetland location or might have no interest in wetlands since they chose other attractions over wetlands. As it was suspected that inadequate sample representativeness might lead to sample risk or the inaccurate measurement of the intention of changing tourism patterns, the questionnaire survey was carried out on tourists visiting the research areas of the two wetlands only.

The sample composition was as follows. In terms of gender, there were 321 males and 230 females. In terms of educational level, 108 respondents had a high school degree (and below), 305 had a bachelor's degree, and 138 had a master's degree (and above). In terms of marital status, 258 respondents were single, and 293 were married. In terms of family status, 240 respondents had children, and 311 had no children. In terms of age, 140 respondents were 30 years old (and below), 161 were 31 to 40 years old, 114 were 41 to 50 years old, 82 were 51 to 60 years old, and 54 were 61 years old and above. In terms of past tour experiences to wetlands, 205 respondents had visited both wetlands in the past, and 346 had not visited them before. In terms of the intention to visit peripheral attractions, 426 respondents would visit peripheral attractions of the wetlands, and 125 would not visit peripheral attractions of the wetlands. In terms of consuming behaviors, 416 respondents would spend money at wetland attractions, and 136 would not spend money at wetland attractions.

In order to confirm the sample representation, this study evaluated the effects of non-response by the wave analysis method. Therefore, non-respondent bias was evaluated by comparing the first batch of backfill data (early respondents) with the second batch of backfill data (late respondents) [49]. According to Armstrong and Overton [49], the key characteristics of early response data and late response data were analyzed by the t-test. There was no significant difference in the age of early and late respondents at a significance level of 5%. Therefore, non-respondent bias in this study was insignificant.

## 4. Empirical Results

### 4.1. Analytical Results of Common Method Variance

When a respondent answered all variables or made measurements, it was easy to produce single source bias and, therefore, that common method variance (CMV) might exist in the study [50]. In terms of prior prevention, this study avoided the generation of CMV by methods such as survey information hiding and reverse item design.

This study carried out the post hoc test of CMV by Harman's single factor analysis [51]. In this study, three factors were generated through factor analysis without rotation on all measurement items, accounting for 64.042% of the cumulative interpretation, while Factor 1 accounted for 28.535% of the variance, which did not exceed the judgment criteria of 50%. Since single factors failed to generate a large variance, the problems caused by the common method variance in this study were not too serious [52].

### 4.2. Descriptive Statistics and Correlation Analysis

The descriptive statistics and Pearson correlation coefficient analysis of research variables are shown in Table 1. There was a significant positive correlation among all variables. Generally, Cronbach's $\alpha$ was used to measure the consistency and stability of the questionnaire, because it was easy to calculate and is a reliability measurement method commonly used in social science research. The higher $\alpha$ is, the greater the correlation is between the items in this factor, and the higher is the consistency. Cronbach's $\alpha$ above 0.7 indicated high reliability, and Cronbach's $\alpha$ below 0.35 indicated low reliability. According to Table 1, Cronbach's $\alpha$ of the reliability in all dimensions was above 0.8, which met the requirements of internal consistency, indicating high reliability of all dimensions in this study.

**Table 1.** Descriptive statistics and correlation analysis.

| | 1 | 2 | 3 | 4 | 5 |
|---|---|---|---|---|---|
| 1. Crowding perception | 1 | | | | |
| 2. Nature-based destination attractiveness | 0.614 *** | 1 | | | |
| 3. Risk perception of COVID-19 | 0.748 *** | 0.536 *** | 1 | | |
| 4. Switching intention to wetland ecotourism | 0.640 *** | 0.512 *** | 0.581 *** | 1 | |
| 5. Wetland ecotourism behaviors | 0.637 *** | 0.751 *** | 0.547 *** | 0.582 *** | 1 |
| Mean | 4.3766 | 04.1393 | 04.4877 | 4.3466 | 4.2187 |
| S.D. | 0.43087 | 0.41805 | 0.49586 | 0.46387 | 0.45131 |
| Cronbach's α | 0.822 | 0.812 | 0.920 | 0.850 | 0.941 |

*** $p < 0.001$, n = 551.

### 4.3. Results of Confirmatory Factor Analysis

The measurement model was tested by confirmatory factor analysis. According to Table 2, all t values of the loading of the measured items in all dimensions were higher than the significance level of 1.96, and the factor loading (λ) of all observed variables for individual latent variables was between 0.60 and 0.96. These values were above the threshold value of 0.45 proposed by Bentler and Wu [53], indicating considerable convergent validity of the scale. The individual item reliability of observed variables was between 0.36 and 0.92. These values were above the threshold value of 0.20 proposed by Bentler and Wu [53], indicating that they met the requirement on the reliability of single variables and all observed variables had reliability. In terms of composite reliability (CR) of the five dimensions, CR was between 0.79 and 0.94. Most past scholars have suggested that the CR of latent variables should be higher than 0.6 [54]. Here, the CR of all dimensions was higher than 0.79, indicating the reliability of all dimensions. The average variance extracted (AVE) of five dimensions was between 0.49 and 0.80. An AVE above 0.36 is a barely acceptable standard [54]. This study was consistent with the views of Bentler and Wu [53] and Fornell and Larcker [54]. The AVE of all dimensions was greater than 0.49, indicating convergent validity of all dimensions.

**Table 2.** Individual item reliability, composite reliability, and average variance extracted.

| Construct | No. of Items | Factor Loading (λ) | Individual Item Reliability (λ²) | t-Value | Composite Reliability (CR) | Average Variance Extracted (AVE) |
|---|---|---|---|---|---|---|
| 1. Crowding perception | 4 | 0.60~0.86 | 0.36~0.74 | 19.16~24.54 | 0.83 | 0.55 |
| 2. Nature-based destination attractiveness | 4 | 0.60~0.80 | 0.36~0.64 | 14.81~21.60 | 0.79 | 0.49 |
| 3. Risk perception of COVID-19 | 4 | 0.82~0.90 | 0.67~0.81 | 23.07~26.86 | 0.92 | 0.74 |
| 4. Switching intention to wetland ecotourism | 3 | 0.66~0.96 | 0.52~0.92 | 16.90~29.03 | 0.87 | 0.70 |
| 5. Wetland ecotourism behaviors | 4 | 0.87~0.93 | 0.76~0.86 | 25.30~27.15 | 0.94 | 0.80 |

### 4.4. Empirical Testing of Moderating Effect of Mooring Factor

In order to deeply understand the influences of push, pull, and mooring on the switching intention to wetland ecotourism and the moderating effect of mooring, hierarchical regression analysis was used in this study for discussion. In Model 1 of Table 3, independent variables, such as crowding perception, risk perception of COVID-19, and nature-based destination attractiveness, were used as the basis to compare with other models. According to the results of the regression analysis, in Model 1, $R^2 = 0.450$, adj-$R^2 = 0.447$, and F value = 149.075. Model 1 reached the significance level of $p < 0.001$ and showed the significantly positive influences of three variables, such as crowding perception (B = 0.415; $p < 0.001$), nature-based destination attractiveness (B = 0.184; $p < 0.001$), and risk perception of COVID-19 (B = 0.191; $p < 0.001$), on the switching intention to wetland ecotourism. Therefore, H1, H2, and H3 are supported.

**Table 3.** Results of regression analysis.

| | Model 1 | | | | Model 2 | | | | Model 3 | | | |
| --- | --- | --- | --- | --- | --- | --- | --- | --- | --- | --- | --- | --- |
| | B | S.E. | t | VIF | B | S.E. | t | VIF | B | S.E. | t | VIF |
| Constant | 0.913 *** | 0.166 | 5.486 | | 0.693 *** | 0.172 | 4.042 | | 0.850 *** | 0.167 | 5.078 | |
| Crowding perception (H1) | 0.415 *** | 0.056 | 7.465 | 2.653 | 0.389 *** | 0.055 | 7.055 | 2.687 | 0.404 *** | 0.055 | 7.279 | 2.669 |
| Risk perception of COVID-19 (H3) | 0.191 *** | 0.045 | 4.214 | 2.322 | 0.237 *** | 0.046 | 5.180 | 2.458 | 0.217 *** | 0.046 | 4.713 | 2.443 |
| Nature-based destination attractiveness (H2) | 0.184 *** | 0.045 | 4.083 | 1.639 | 0.206 *** | 0.045 | 4.618 | 1.661 | 0.179 *** | 0.045 | 3.983 | 1.643 |
| Risk perception of COVID-19 × Crowding perception (H4) | | | | | 0.239 *** | 0.055 | 4.326 | 1.116 | | | | |
| Risk perception of COVID-19 × Nature-based destination attractiveness (H5) | | | | | | | | | 0.123 ** | 0.047 | 2.617 | 1.062 |
| $R^2$ | | 0.450 | | | | 0.468 | | | | 0.457 | | |
| $Adj - R^2$ | | 0.447 | | | | 0.464 | | | | 0.453 | | |
| F | | 149.075 *** | | | | 120.106 *** | | | | 114.714 *** | | |
| $\Delta R^2$ | | | | | | 0.018 *** | | | | 0.007 *** | | |
| ΔF | | | | | | 18.716 *** | | | | 6.849 ** | | |

B is an unstandardized coefficient. ** $p < 0.01$, *** $p < 0.001$, n = 551.

For the treatment of multicollinearity, according to the method of Aiken and West [55], the means were subtracted from the main variables for centering to avoid multicollinearity. The first interaction term was assessed in Model 2. Both $\Delta R^2$ and ΔF of Model 2 reached a significance level of $p < 0.001$, indicating that the addition of the first interaction variable improved the explanatory power of Model 2. Model 2 showed the significantly positive influences of risk perception of COVID-19 × crowding perception (B = 0.239; $p < 0.001$) on the switching intention to wetland ecotourism.

The second interaction term was assessed in Model 3. Both $\Delta R^2$ and ΔF of Model 3 reached a significance level of $p < 0.001$, indicating that the addition of the second interaction variable improved the explanatory power of Model 3. Model 3 showed significantly positive influences of the risk perception of COVID-19 × nature-based destination attractiveness (B = 0.123; $p < 0.01$) on the switching intention to wetland ecotourism. However, whether mooring had a moderating effect could only be known by drawing a moderating diagram. The moderating diagram is analyzed in the next section.

### 4.5. Analytical Results of Interaction Plot

Based on the method of Aiken and West [55], this study added or subtracted a standard deviation from each of the means of variables (crowding perception, nature-based destination attractiveness, and risk perception of COVID-19) and each of the variables. The results were divided into two groups with high and low scores and then loaded into the regression model to draw the interactions for the purpose of further explaining the form of interaction among variables. In the two figures, the solid lines show the high-risk perception of COVID-19, and the dotted lines show the low-risk perception of COVID-19.

In Figure 4 the horizontal axis shows the degree of change in crowding perception, and the vertical axis shows the degree of change in switching intention to wetland ecotourism. The two lines in Figure 4 show obvious intersecting points, and the phenomenon of interaction is known as disordinal interaction, indicating the moderating effect of risk perception of COVID-19. According to the slopes of the two lines, the line of high-risk perception of COVID-19 is steeper than that of low-risk perception of COVID-19, therefore, it could be concluded that high-risk perception of COVID-19 was more effective in enhancing the consistency between the crowding perception and switching intention to wetland ecotourism (positive relationship) than the low-risk perception of COVID-19. The interaction diagram shows in the case of a high degree of crowding perception that tourists with a high-risk perception of COVID-19 had a higher intention to switch to wetland ecotourism than that of those with a low-risk perception of COVID-19. In other words, the degree of switching intention to wetland ecotourism varied with the risk perception of COVID-19. Therefore, H4 is supported.

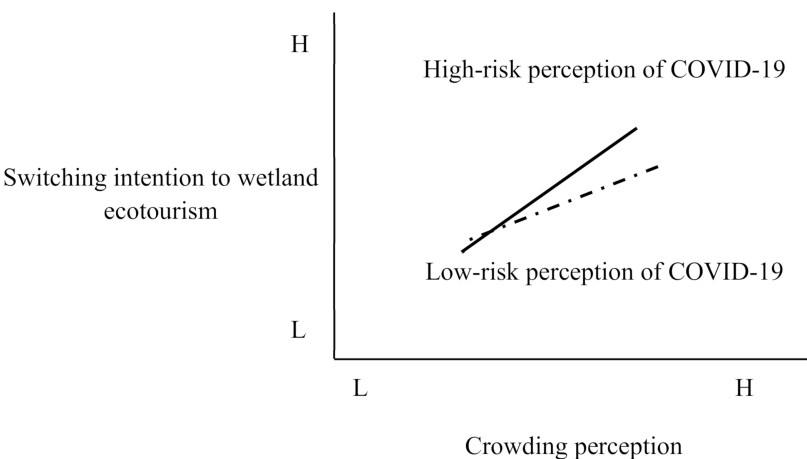

**Figure 4.** Moderation effect of perception of COVID-19 on the relation between crowding perception and switching intention to wetland ecotourism.

In Figure 5 the horizontal axis shows the degree of change in nature-based destination attractiveness, and the vertical axis shows the degree of switching intention to wetland ecotourism. However, the two lines in Figure 5 tend to be parallel with no cross point, which shows that risk perception of COVID-19 had no moderating effect on the relationship between the nature-based destination attractiveness and switching intention to wetland ecotourism. The interaction diagram shows that the degree of switching intention to wetland ecotourism did not vary with the risk perception of COVID-19. Therefore, H5 is not supported.

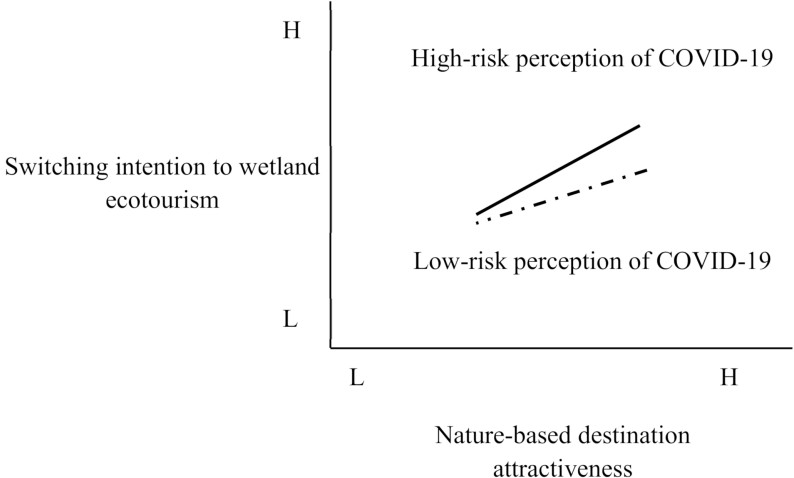

**Figure 5.** Moderation effect of perception of COVID-19 on the relation between nature-based destination attractiveness and switching intention to wetland ecotourism.

### 4.6. Influences of Switching Intention to Wetland Ecotourism on Wetland Ecotourism Behaviors

Model 4 in Table 4 shows the influences of switching intention to wetland ecotourism on wetland ecotourism behaviors. According to the results of the regression analysis, with $R^2 = 0.338$, adj-$R^2 = 0.337$, and F value = 280.799, Model 4 reached a significance level of $p < 0.001$. Model 4 showed the significantly positive influences of switching intention to wetland ecotourism (B = 0.566; $p < 0.001$) on wetland ecotourism behaviors. Therefore, H6 is supported.

**Table 4.** Results of regression analysis.

| | Model 4 | | | |
| --- | --- | --- | --- | --- |
| | **B** | **S.E.** | **t** | **VIF** |
| Constant | 1.759 *** | 0.148 | 11.912 | |
| Switching intention to wetland ecotourism (H6) | 0.566 *** | 0.034 | 16.757 | 1.000 |
| $R^2$ | | 0.338 | | |
| $Adj - R^2$ | | 0.337 | | |
| F | | 280.799 *** | | |

B is an unstandardized coefficient. *** $p < 0.001$, n = 551.

## 5. Conclusions and Implications

### 5.1. Discussion

The literature review found only a few studies exploring tourism switch intention or tourism behaviors using the PPM model [23–26]. In particular, studies on tourists' intention to switch to wetland ecotourism during the COVID-19 pandemic in 2020 were rare. Therefore, this study used the PPM model to explain tourists' intentions to forgo urban tourism and switch to wetland ecotourism, in addition to their behaviors toward wetland ecotourism. With the COVID-19 pandemic, the thoughts and behaviors of travelers have become more unpredictable. Nevertheless, this study successfully explained these behaviors by employing the three forces of the PPM model while attempting to find answers to complex behavioral predictions. This study is expected to bring innovations to this topic and research as a basis for subsequent research.

### 5.2. Conclusions

The empirical results showed that the greater the push (namely, crowding perception), pull (namely, nature-based destination attractiveness), and mooring (namely, risk perception of COVID-19) that tourists perceived were, the stronger was the switching intention to wetland ecotourism. The risk perception of COVID-19 positively moderated the relationship between crowding perception and tourists' switching intention to wetland ecotourism, but did not moderate the relationship between nature-based destination attractiveness and tourists switching intention to wetland ecotourism. This might be because regardless of the influences of COVID-19, wetland ecological attractions were already attractive enough to tourists and had enough incentives to change their tour intention to visit wetland attractions. Subsequently, this indicated the insignificant moderating effect of the risk perception of COVID-19. According to the empirical results, tourists' switching intention to wetland ecotourism had significantly positive influences on wetland ecotourism behaviors, which was the same as previous studies on the relationship between intention and behavior [41–43].

### 5.3. Implications

With the spread and impact of COVID-19, the global tourism market has been greatly affected. Foreign tourists cannot enter Taiwan, and its citizens cannot leave Taiwan for travel. Therefore, Taiwanese have begun to dislike urban tours and have switched to attractions in Taiwan, especially outdoor leisure activities. Taiwan has many wetland attractions with ecological diversity, including two world-class wetlands, 42 important national wetlands, and 41 temporarily important local wetlands. However, ecotourism is not as popular as other forms of tours in Taiwan. While there are sufficient ecological interpreters and environmental protection education at ecological attractions, the shortage of ecological interpreters in many ecotourism attractions in Taiwan leads to a lack of ecological knowledge in the process of ecotourism. This lack of ecological knowledge makes Taiwanese feel that ecotourism is not interesting, and thus affects their intention to visit and revisit ecological attractions. Under the current COVID-19 pandemic, Taiwanese have switched to outdoor ecotourism attractions for fear of crowding in urban attractions. County and

municipal tour authorities in Taiwan can thus re-examine their wetland ecotourism policies now to develop supporting measures for wetland ecotourism, to encourage people to visit wetland ecotourism attractions and peripheral attractions for in-depth eco-cultural tours, and to promote the development of the local ecotourism economy.

*5.4. Future Research Directions*

This study proposes several future research directions as a reference for subsequent researchers. First, the selection of variables (e.g., push factors, pull factors, and mooring factors) requires further discussion. Different research themes can use different measurement variables according to the context. It is suggested that subsequent researchers use different variables to explain the influencing factors of the intentions for switching to wetland ecotourism under the framework of this study. Second, with diverse forms of tourism available, it is suggested that subsequent researchers use the same framework to explore the different types of tourism. Third, different medical environments and levels in different countries and regions also lead to differences in the recovery status under the COVID-19 pandemic. Therefore, it is suggested that subsequent researchers conduct research and comparisons for different countries and regions under the same framework. Furthermore, this study posits that the PPM model can be generally used for studies in broad fields. Moreover, it is expected that research will find various findings if subsequent researchers conduct studies with different fields, variables, and objects.

**Author Contributions:** Conceptualization, Y.-W.W., T.-H.L. and S.-P.Y.; Data curation, T.-H.L.; Investigation, T.-H.L.; Methodology, T.-H.L. and H.-C.H.; Supervision, Y.-W.W.; Writing—Original draft, Y.-W.W., T.-H.L. and H.-C.H.; Writing—Review & editing, S.-P.Y. All authors have read and agreed to the published version of the manuscript.

**Funding:** This research received no external funding.

**Conflicts of Interest:** The authors declare no conflict of interest.

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
