# Peer review of "Switching Intention and Behaviors to Wetland Ecotourism after the COVID-19 Pandemic: The Perspective of Push-Pull-Mooring Model"

_sustainability, doi:10.3390/su14106198_

Round 1

Reviewer 1 Report

The author says that this study is a study on 'Switching Intention to Ecotourism'. However, the actual research is limited to wetland tourism (wetland attack) This is more evident in questions such as P.7, Mooring Effect of L.271~282, P.7, and Switching extension to ecotourism of L287~291. (Question is limited to the idea of a wetland attack.) Therefore, it is necessary to clearly limit the title and the entire content of the thesis.
As respondents target people who visit wetlands.
So, It is necessary to further clarify the purpose and scope of this study, and to strengthen the validity of the question contents and respondents.

Author Response

Reviewer #1

Response:

First of all, we appreciate your comments and suggestions. Your comments and suggestions make a valuable contribution to this paper. If this paper is accepted by the Sustainability, we definitely ascribe the achievement to your assistance. We have responded to your comments and suggestions in sequence as follows.

Major comments:

The author says that this study is a study on 'Switching Intention to Ecotourism'. However, the actual research is limited to wetland tourism (wetland attack) This is more evident in questions such as P.7, Mooring Effect of L.271~282, P.7, and Switching extension to ecotourism of L287~291. (Question is limited to the idea of a wetland attack.) Therefore, it is necessary to clearly limit the title and the entire content of the thesis.

As respondents target people who visit wetlands.

So, It is necessary to further clarify the purpose and scope of this study, and to strengthen the validity of the question contents and respondents.

Response:

Thank you for your comments. We have revised the title as suggested. Accordingly, the original title, “Switching Intention and Behaviors to Ecotourism after the COVID-19 Pandemic: The Perspective of Push-Pull-Mooring Model”, was duly changed to “Switching Intention and Behaviors to Wetland Ecotourism after the COVID-19 Pandemic: The Perspective of Push-Pull-Mooring Model”.

In the text, we accordingly used the terms “wetland ecotourism” and “wetland ecotourism behaviors” instead of “ecotourism” and “ecotourism behaviors” to conform to the research content, purpose, and scope.

Reviewer 2 Report

Many thanks for giving me the chance to review such a paper. This study intends to explain the influencing factors of tourists’ switching intention to ecotourism after the COVID-19 11 pandemic.

First, I would specify and highlight specific theories that have been applied when examining these issues. Several of the research articles cited in your theoretical framework highlight applicable theories, and it would be beneficial to highlight these and indicate which are most valuable for your research.

A map of the geographical area studied would add value to the study.

In the introductory section and the theoretical framework, there is a lack of tourism-related issues that should at least be mentioned: rural tourism, overtourism, seasonality, technological innovation, etc. as factors to be taken into account in future research:

Medina, R. M. P., Martín, J. M. M., Martínez, J. M. G., & Azevedo, P. S. (2022). Analysis of the role of innovation and efficiency in coastal destinations affected by tourism seasonality. Journal of Innovation & Knowledge7(1), 100163.

Fernández, J. A. S., Martínez, J. M. G., & Martín, J. M. M. (2022). An analysis of the competitiveness of the tourism industry in a context of economic recovery following the COVID19 pandemic. Technological Forecasting and Social Change, 174, 121301.

Martín, J. M. M., & Fernández, J. A. S. (2022). The effects of technological improvements in the train network on tourism sustainability. An approach focused on seasonality. Sustainable Technology and Entrepreneurship, 1(1), 100005.

Second, it would be helpful to separate your results and discussion section, emphasizing the discussion to tie the results back to your original research questions.

To show the results of statistical methods is not enough, author(s) has(ve) to explain what these results mean from the view of the investigation. For example: are they acceptable, and can the investigation be continued based on them? Are there problematic relationships, or some of them must be neglected? Etc.

Finally, you should develop your conclusion section more fully by including future research directions. The paper contributes significantly to this field, and you should indicate in what area this should go. This may consist of looking at other issues in the future such as international trade levels with other countries and the power/dependence influence from other countries to conform to specific standards, level of education within a country, levels of manufacturing vs. knowledge-based industries within a country, or even age of the government or economy. Several areas may play an important role, and you should offer ideas for others who will build on your work.

English proficiency can be further improved.

Kind regards

Author Response

Reviewer #2

Response:

First of all, we appreciate your comments and suggestions. Your comments and suggestions make a valuable contribution to this paper. If this paper is accepted by the Sustainability, we definitely ascribe the achievement to your assistance. We have responded to your comments and suggestions in sequence as follows.

Major comments:

  1. First, I would specify and highlight specific theories that have been applied when examining these issues. Several of the research articles cited in your theoretical framework highlight applicable theories, and it would be beneficial to highlight these and indicate which are most valuable for your research.

A map of the geographical area studied would add value to the study.

Response:

Thank you for your comments. We have added geographic maps of the two wetlands in Section 3.3 of the revised paper, as follows:

The geographic scope of Sihcao Important Wetland is shown in Figure 2. The geographic scope of Zengwun Estuary Important Wetland is shown in Figure 3.

  1. In the introductory section and the theoretical framework, there is a lack of tourism-related issues that should at least be mentioned: rural tourism, overtourism, seasonality, technological innovation, etc. as factors to be taken into account in future research:

Medina, R.M.P., Martín, J.M.M., Martínez, J.M.G., & Azevedo, P.S. (2022). Analysis of the role of innovation and efficiency in coastal destinations affected by tourism seasonality. Journal of Innovation & Knowledge, 7(1), 100163.

Fernández, J.A.S., Martínez, J.M.G., & Martín, J.M.M. (2022). An analysis of the competitiveness of the tourism industry in a context of economic recovery following the COVID19 pandemic. Technological Forecasting and Social Change, 174, 121301.

Martín, J.M.M., & Fernández, J.A.S. (2022). The effects of technological improvements in the train network on tourism sustainability. An approach focused on seasonality. Sustainable Technology and Entrepreneurship, 1(1), 100005.

Response:

Thank you for your comments. We have included your provided references in the revised paper’s “Introduction” section, as follows:

Paragraph 1.

Especially after the COVID-19 outbreak, commercial tourism suffered greatly (Fernández, Martínez, & Martín, 2022), as many people no longer consider crowded urban tours their first option for travel destinations.

Paragraph 2.

Seasonality is an important determinant of tourism competitiveness (Martín, & Fernández, 2022; Medina, Martín, Martínez, & Azevedo, 2022). Meanwhile, wetlands show different natural features and environmental ecology in different seasons, which could attract tourists.

References

Fernández, J.A.S.; Martínez, J.M.G.; Martín, J.M.M. An analysis of the competitiveness of the tourism industry in a context of economic recovery following the COVID19 pandemic. Technol. Forecast. Soc. Chang. 2022, 174, 121301.

Martín, J.M.M.; Fernández, J.A.S. The effects of technological improvements in the train network on tourism sustainability. An approach focused on seasonality. Sustainable Technology and Entrepreneurship 2022, 1(1), 100005.

Medina, R.M.P.; Martín, J.M.M.; Martínez, J.M.G.; Azevedo, P.S. Analysis of the role of innovation and efficiency in coastal destinations affected by tourism seasonality. J. Innov. Knowl. 2022, 7(1), 100163.

  1. Second, it would be helpful to separate your results and discussion section, emphasizing the discussion to tie the results back to your original research questions

Response:

Thank you for your comments. We have divided “5. Conclusions and Implications” into four subsections in the revised paper: “5.1. Discussion”, “5.2. Conclusions”, “5.3. Implications”, and “5.4. Future Research Directions”. Below is the following revision for subsection “5.1. Discussion”:

5.1. Discussion

The literature review found only a few studies exploring tourism switch intention or tourism behaviors by the PPM model [17-20]. In particular, studies on tourists’ intention to switch to wetland ecotourism during the COVID-19 pandemic in 2020 are rare. Therefore, this study used the PPM model to explain tourists’ intentions to give up urban tourism and switch to wetland ecotourism, as well as their behaviors toward wetland ecotourism. With the COVID-19 pandemic, the thoughts and behaviors of travelers have become more unpredictable. Nevertheless, this study successfully explained these behaviors by employing the three forces of the PPM model while attempting to find answers to complex behavioral predictions. This study is expected to bring innovations to this topic and research as a basis for subsequent research.

  1. To show the results of statistical methods is not enough, author(s) has(ve) to explain what these results mean from the view of the investigation. For example: are they acceptable, and can the investigation be continued based on them? Are there problematic relationships, or some of them must be neglected? Etc.

Response:

Thank you for your comments. We have added the subsections “5.1. Discussion” and “5.4. Future Research Directions” to “5. Conclusions and Implications”.

  1. Finally, you should develop your conclusion section more fully by including future research directions. The paper contributes significantly to this field, and you should indicate in what area this should go. This may consist of looking at other issues in the future such as international trade levels with other countries and the power/dependence influence from other countries to conform to specific standards, level of education within a country, levels of manufacturing vs. knowledge-based industries within a country, or even age of the government or economy. Several areas may play an important role, and you should offer ideas for others who will build on your work.

Response:

Thank you for your comments. We have added the subsection “5.4. Future Research Directions” to “5. Conclusions and Implications”. Below is the following revision for the subsection:

5.4. Future Research Directions

This study proposed several future research directions as a reference for subsequent researchers. First, the selection of variables (e.g., push factors, pull factors, and mooring factors) requires further discussion. Different research themes can use different measurement variables according to the context. It is suggested that subsequent researchers use different variables to explain the influencing factors of the intentions for switching to wetland ecotourism under the framework of this study. Second, with the diversity in the forms of tourism, it is suggested that subsequent researchers use the same framework to explore the different types of tourism. Third, different medical environments and levels in different countries and regions also lead to differences in the recovery status under the COVID-19 pandemic. Therefore, it is suggested that subsequent researchers conduct research and comparisons for different countries and regions under the same framework. Furthermore, this study posited that the PPM model could be used generally for studies in broad fields. Moreover, it is expected that research will find various findings if subsequent researchers conduct studies with different fields, variables, and objects.

  1. English proficiency can be further improved.

Response:

Thank you for your comments. The English translation of our paper has been polished by a professional translation agency and a native English-speaking editor before submission. We believe that it meets the quality standards and requirements of journals.

Reviewer 3 Report

This paper is a quantitative research article, through the examination of conceptual framework, the proposed questions have explored.

This paper has presented an interesting research direction; some comments are following:

Section 2 is more preferred as literature review; but in the section has no any background can find or talk about the ecotourism.

In doing so, why the Figure 1 (conceptual framework) has a relationship with ecotourism behavior. Although H6 has discussed and received results.

In the Section 5, what is the future works, I cannot find it out.

Author Response

Reviewer #3

Response:

First of all, we appreciate your comments and suggestions. Your comments and suggestions make a valuable contribution to this paper. If this paper is accepted by the Sustainability, we definitely ascribe the achievement to your assistance. We have responded to your comments and suggestions in sequence as follows.

Major comments:

  1. Section 2 is more preferred as literature review; but in the section has no any background can find or talk about the ecotourism.

In doing so, why the Figure 1 (conceptual framework) has a relationship with ecotourism behavior. Although H6 has discussed and received results.

Response:

Thanks for your comments. We have divided “2. Theory and Hypotheses” into five subsections in the revised paper: “2.1. Migration Theory of Push-Pull-Mooring Model (PPM)”, “2.2. Ecotourism”, “2.3. Ecotourism Behaviors”, “2.4. Push-Pull Mooring Model and Switching Behaviors”, and “2.5. Hypotheses”.

In the original subsection “2.2. Ecotourism Behaviors”, the first paragraph introduces the literature on ecotourism, and the second paragraph introduces the literature on ecotourism behaviors. Meanwhile, in the revised paper, the subsection “2.2. Ecotourism Behaviors” has been divided into two subsections: “2.2. Ecotourism” and “2.3. Ecotourism Behaviors”. Below are the following revisions:

2.2. Ecotourism

…Chiu, Lee, and Chen (2014) argued that ecotourism attaches importance to the sustainable development of the environment, and environmentally responsible behaviors belong to a kind of environmental protection mechanism. Additionally, Cai, Liu, and Zhang (2019) pointed out that ecotourism refers to the special utilization of natural areas without disturbance and pollution, where tourists can enjoy natural activities, learn to protect local resources, and give back to community development to achieve the ultimate goal of sustainable management.

2.3. Ecotourism Behaviors

Ecotourism behaviors refer to environmentally responsible behaviors. In the context of ecotourism, when tourists understand the impact of their actions on the environment and adhere to the norms of ecological attractions, they will maintain environmentally responsible behaviors (Puhakka, 2011)…

References

Chiu, Y.T.H.; Lee, W.I.; Chen, T.H. Environmentally responsible behavior in ecotourism: Antecedents and implications. Tourism Manage. 2014, 40, 321-329.

Cai, W.M.; Liu, X.; Zhang, W. Effects of value and attitude on environment behaviour in ecotourism. J. Environ. Prot. Ecol. 2019, 20(A), 16-22.

Puhakka, R. Environmental concern and responsibility among nature tourists in Oulanka Pan park, Finland. Scand. J. Hosp. Tour. 2011, 11(1), 76-96.

  1. In the Section 5, what is the future works, I cannot find it out.

Thank you for your comments. We have added the subsection “5.4. Future Research Directions” to “5. Conclusions and Implications”. Below is the following revision for the subsection:

5.4. Future Research Directions

This study proposed several future research directions as a reference for subsequent researchers. First, the selection of variables (e.g., push factors, pull factors, and mooring factors) requires further discussion. Different research themes can use different measurement variables according to the context. It is suggested that subsequent researchers use different variables to explain the influencing factors of the intentions for switching to wetland ecotourism under the framework of this study. Second, with the diversity in the forms of tourism, it is suggested that subsequent researchers use the same framework to explore the different types of tourism. Third, different medical environments and levels in different countries and regions also lead to differences in the recovery status under the COVID-19 pandemic. Therefore, it is suggested that subsequent researchers conduct research and comparisons for different countries and regions under the same framework. Furthermore, this study posited that the PPM model could be used generally for studies in broad fields. Moreover, it is expected that research will find various findings if subsequent researchers conduct studies with different fields, variables, and objects.

Reviewer 4 Report

The authors prepared an interesting application of the Push – Pull – Mooring model in a study where they linked ecotourism to the Covid situation. Methodologically sovereign, with appropriate sampling and testing of the questionnaire. The findings are interesting and instructive. Only the mostly older literature used in the article deviates slightly from the otherwise excellent image. References are double numbered.

Author Response

Reviewer #4

Response:

First of all, we appreciate your comments and suggestions. Your comments and suggestions make a valuable contribution to this paper. If this paper is accepted by the Sustainability, we definitely ascribe the achievement to your assistance. We have responded to your comments and suggestions in sequence as follows.

Major comments:

The authors prepared an interesting application of the Push – Pull – Mooring model in a study where they linked ecotourism to the Covid situation. Methodologically sovereign, with appropriate sampling and testing of the questionnaire. The findings are interesting and instructive. Only the mostly older literature used in the article deviates slightly from the otherwise excellent image. References are double numbered.

Response:

Thank you for your comments. We have included three references in the “Introduction” section of the revised paper. Noteworthily, it was found that the journal’s editorial department is responsible for the double-numbered references you mentioned.

References

Fernández, J.A.S.; Martínez, J.M.G.; Martín, J.M.M. An analysis of the competitiveness of the tourism industry in a context of economic recovery following the COVID19 pandemic. Technol. Forecast. Soc. Chang. 2022, 174, 121301.

Martín, J.M.M.; Fernández, J.A.S. The effects of technological improvements in the train network on tourism sustainability. An approach focused on seasonality. Sustainable Technology and Entrepreneurship 2022, 1(1), 100005.

Medina, R.M.P.; Martín, J.M.M.; Martínez, J.M.G.; Azevedo, P.S. Analysis of the role of innovation and efficiency in coastal destinations affected by tourism seasonality. J. Innov. Knowl. 2022, 7(1), 100163.

Round 2

Reviewer 1 Report

The researcher selected wetland tourism as the theme among ecotourism. Therefore, it is necessary to strengthen the importance of wetland tourism in the content by comparing and citing previous studies of various types of ecotourism.

The author conducted a survey of wetland visitors. A supplementary explanation is needed for the justification of the sample group (whether non-wetland visitors or visitors from other regions have been considered).

It is necessary to reinforce the importance of the study by reinforcing the academic, policy, and industrial implications of this study.

Author Response

Reviewer #1

Response:

First of all, we appreciate your comments and suggestions. Your comments and suggestions make a valuable contribution to this paper. If this paper is accepted by the Sustainability, we definitely ascribe the achievement to your assistance. We have responded to your comments and suggestions in sequence as follows.

Major comments:

The researcher selected wetland tourism as the theme among ecotourism. Therefore, it is necessary to strengthen the importance of wetland tourism in the content by comparing and citing previous studies of various types of ecotourism.

The author conducted a survey of wetland visitors. A supplementary explanation is needed for the justification of the sample group (whether non-wetland visitors or visitors from other regions have been considered).

It is necessary to reinforce the importance of the study by reinforcing the academic, policy, and industrial implications of this study.

Response:

Thank you for your comments. We have made corrections and explanations in Section 3.4 Sample and Procedure in the revised paper. The corrections are as follows:

The subjects of this study were tourists visiting two wetland areas, while tourists visiting non-wetland attractions or attractions in other areas were not included. The intentions and behaviors of tourists converting their normal travel destinations to wetland tourism, which were the topics explored by this study, were the major concerns in the research design. This study focused on the intentions and behaviors of this conversion. Tourists visiting non-wetland attractions or attractions in other areas might not know the wetland location or might have no interest in wetlands since they chose other attractions over wetlands. As it was suspected that inadequate sample representativeness might lead to sample risk or the inaccurate measurement of the intention of changing tourism patterns of this study, the questionnaire survey has been carried out on tourists visiting the research areas of the two wetlands only.

Reviewer 2 Report

Congratulations, the modifications and suggestions have been made to improve the scientific robustness of your article. Good work.

Author Response

Reviewer #2

Response:

Major comments:

Congratulations, the modifications and suggestions have been made to improve the scientific robustness of your article. Good work.

Response:

First of all, we appreciate your comments and suggestions. Your comments and suggestions make a valuable contribution to this paper. If this paper is accepted by the Sustainability, we definitely ascribe the achievement to your assistance.

Reviewer 3 Report

I am so happy to see the revised version, thanks authors. This is my second time to read this paper, I think that this renewed version is suitable to be published on the journal.

My opinion is this paper has according to my suggestions, although those opinions may not good enough but it still can make the paper more clearly.

As the renewed version, it provides more information for readers such a map to show what they do for the concerned problem, I think that is good idea.  

Author Response

Reviewer #3

Response:

First of all, we appreciate your comments and suggestions. Your comments and suggestions make a valuable contribution to this paper. If this paper is accepted by the Sustainability, we definitely ascribe the achievement to your assistance. We have responded to your comments and suggestions in sequence as follows.

Major comments:

I am so happy to see the revised version, thanks authors. This is my second time to read this paper, I think that this renewed version is suitable to be published on the journal.

My opinion is this paper has according to my suggestions, although those opinions may not good enough but it still can make the paper more clearly.

As the renewed version, it provides more information for readers such a map to show what they do for the concerned problem, I think that is good idea. 

Response:

Thank you for your comments. We have made corrections and explanations in Section 3.3 Study Area in the revised paper. Moreover, two scope maps of the two wetland areas (Sihcao Important Wetland and Zengwun Estuary Important Wetland), as shown in Figure 2 and Figure 3, respectively, were added in the paper, in order that readers could better understand the locations of the two wetlands and the research areas of this study, and thus, further understand the research topic.